# A Risk Prediction Model and Risk Score of SARS-CoV-2 Infection Following Healthcare-Related Exposure

**DOI:** 10.3390/tropicalmed7090248

**Published:** 2022-09-14

**Authors:** Kantarida Sripanidkulchai, Pinyo Rattanaumpawan, Winai Ratanasuwan, Nasikarn Angkasekwinai, Susan Assanasen, Peerawong Werarak, Oranich Navanukroh, Phatharajit Phatharodom, Teerapong Tocharoenchok

**Affiliations:** 1Department of Preventive and Social Medicine, Faculty of Medicine Siriraj Hospital, Mahidol University, Bangkok 10700, Thailand; 2Division of Infectious Diseases and Tropical Medicine, Department of Medicine, Faculty of Medicine Siriraj Hospital, Mahidol University, Bangkok 10700, Thailand; 3Division of Cardiothoracic Surgery, Department of Surgery, Faculty of Medicine Siriraj Hospital, Mahidol University, Bangkok 10700, Thailand

**Keywords:** COVID-19, SARS-CoV-2, occupational exposure, risk factors, personal protective equipment

## Abstract

Hospital workers are at high risk of contact with COVID-19 patients. Currently, there is no evidence-based, comprehensive risk assessment tool for healthcare-related exposure; so, we aimed to identify independent factors related to COVID-19 infection in hospital workers following workplace exposure(s) and construct a risk prediction model. We analyzed the COVID-19 contact tracing dataset from 15 July to 31 December 2021 using multiple logistic regression analysis, considering exposure details, demographics, and vaccination history. Of 7146 included exposures to confirmed COVID-19 patients, 229 (4.2%) had subsequently tested positive via RT-PCR. Independent risk factors for a positive test were having symptoms (adjusted odds ratio 4.94, 95%CI 3.83–6.39), participating in an unprotected aerosol-generating procedure (aOR 2.87, 1.66–4.96), duration of exposure >15 min (aOR 2.52, 1.82–3.49), personnel who did not wear a mask (aOR 2.49, 1.75–3.54), exposure to aerodigestive secretion (aOR 1.5, 1.03–2.17), index patient not wearing a mask (aOR 1.44, 1.01–2.07), and exposure distance <1 m without eye protection (aOR 1.39, 1.02–1.89). High-potency vaccines and high levels of education protected against infection. A risk model and scoring system with good discrimination power were built. Having symptoms, unprotected exposure, lower education level, and receiving low potency vaccines increased the risk of laboratory-confirmed COVID-19 following healthcare-related exposure events.

## 1. Introduction

Healthcare workers are at high risk for exposure to COVID-19, both in the community and in the workplace when caring for patients [1]. Infection prevention and control practices are recommended for all hospital workers and include the use of personal protective equipment, physical distancing, source control measures, immunization, and post-exposure management [2]. The early assessment of risk and prompt management are important to protect the health and safety of personnel to prevent in-hospital transmission [3]. On the other hand, the isolation and quarantine associated with COVID-19 that are required of health workers place additional strain on healthcare services during periods of high demand. The individualized estimation of the infection risk of certain exposure of health workers is needed to guide optimal prevention and response strategies.

The exposure risk assessment and management system is currently mainly based on expert opinion, because only a few studies have addressed this problem, and there is the significant heterogeneity of operational definitions for variables that influence exposure risk, such as the measurement of contact duration, distance, the use of a face mask versus a respirator with eye protection, and differing vaccine regimens and efficacies [4,5,6,7,8,9]. Further, most COVID-19 healthcare exposure studies categorized exposure risk using multiple measures in combination (without complete details of individual exposure) and were conducted during periods when less contagious variants were circulating and different vaccine products and regimens were employed [9,10,11].

In the third quarter of 2021, Siriraj Hospital, a 2300-bed referral center in Bangkok with more than 16,000 employees, conducted more than 200 SARS-CoV-2 genetic tests per day for its personnel. Adapted from USCDC, WHO, European and Thailand public health interim guidelines, the hospital risk assessment and management system classified the risk of exposure and recommended appropriate testing times, work restrictions, and quarantine for those who were exposed to confirmed patients with COVID-19 [12,13,14,15,16]. Independent factors associated with COVID-19 infection could be identified using the large and detailed exposure dataset, demographic data, vaccination history, and complete entry and exit test status.

The objectives of this study are to identify independent factors associated with SARS-CoV-2 infection detected via RT-PCR in hospital workers following exposure(s) to confirmed positive patients and to build an evidence-based quantitative risk model and risk score for healthcare-related exposure.

## 2. Materials and Methods

### 2.1. Study Design, Setting, and Protocol

This study is a retrospective cohort analysis. From July 2021 to January 2022, during the increase in the number of cases of COVID-19 caused by the Delta variant, the hospital implemented a contact tracing and risk evaluation system based on exposure characteristics and immunization status to guide risk-specific SARS-CoV-2 tests, work restriction, and quarantine recommendations (Appendix A). Hospital workers who had been exposed to a confirmed case within the contagious period or had any symptoms related to SARS-CoV-2 (Appendix B) were evaluated as per hospital guidelines.

### 2.2. Data Collection and Preparation

Data collection was completed by exposed hospital workers or their representatives directly into a computer spreadsheet (infected person, worker identification, event details, symptoms, and immunization record). Completeness and accuracy were validated using mandatory field entry, data validation, and logic checks with feedback confirmation by responsible infection control officers. If personnel had multiple exposures to the same index person, the risk would be assigned to the highest risk event, and recommendations would be arranged according to the latest significant exposure. The classification of exposure risk (high, moderate, low or insignificant—based on the characteristics of exposure and the use of personal protective equipment (PPE) according to the consensus of the experts of the hospital detailed in Appendix A) and the recommendation were assigned by infectious disease specialists with the aid of software developed by the hospital. This exposure risk category was not introduced directly to the logistic regression model as all individual exposure criteria had already been included.

The variables of interest that were not included in the initial dataset (age, gender, education, and SARS-CoV-2 test results) and those subject to recall errors (immunization record) were provided by the hospital informatics and data innovation center. Missing and conflicting data were manually imputed based on available electronic hospital records.

### 2.3. Study Definition

#### 2.3.1. Vaccine Formula and Potency Grouping

COVID-19 vaccination at least 14 days before exposure was considered to exert a full protective effect and was defined as the completion of the last dose. Due to the wide variety of vaccine combinations among Thai health workers [17], we classified all combination states into three distinct potency groups according to criteria adapted from Thai COVID-19 vaccination guidelines for a booster shot from the Ministry of Public Health in December 2021 (Appendix A) [18,19]. Low-potency combinations included any number of doses of an inactivated vaccine product, or a single dose of any other product (viral vector or mRNA). Moderate-potency combinations included two or more doses of an inactivated vaccine and at least one dose of either a viral vector product or an mRNA product. High-potency combinations included any dose of an inactivated product with at least one dose of viral vector product plus one dose of mRNA platform, or at least two doses of mRNA platform.

#### 2.3.2. Laboratory Analysis and Case Definition

COVID-19 was diagnosed via SARS-CoV-2 genetic detection from respiratory samples using a real-time RT-PCR test, Allplex™ 2019-nCoV Assay (Seegene^®^, Seoul, Korea). The cycle threshold of <40 for the E and N gene and <42 for the RdRp gene was considered positive. To resolve the discrepancies between different genes tested, infectious disease specialists would define the status of the case based on their history and subsequent test(s).

### 2.4. Statistical Analysis

Continuous variables were reported as means with standard deviation and medians with interquartile range, while categorical data were reported using frequencies and percentages. The variables between groups were compared using the independent sample T test or Pearson’s chi-square test (or nonparametric equivalents where appropriate), with statistical significance defined as a *p* value less than 0.05. Using multiple logistic regression, all variables with a *p* value less than 0.25 from univariate pre-screening entered the model provided they were present in at least 1% of the sample. Using the stepwise multivariate analysis, the variables that did not contribute to the model were eliminated either by exclusion or collapse to another category, whichever yielded maximal discrimination power from the ROC curve analysis. An additive risk score of predicted probability of COVID-19 infection was developed with coefficients from the final model (Appendix C). Model fit was accessed using the Hosmer and Lemeshow test. The logistic exposure risk calculator was built and is available at https://bit.ly/3uEi4W2 (accessed on 15 May 2022). All analyses were performed using SPSS™ software version 26.0 (IBM Corporation, Armonk, NY, USA) and Microsoft Excel™ software version 2203 (Microsoft Corporation, Redmond, WA, USA).

## 3. Results

The study flow diagram is illustrated in Figure 1. From 15 July to 31 December 2021, more than 19,000 hospital workers exposed to confirmed SARS-CoV-2 patients or who had symptoms related to COVID-19 were reported to infectious disease specialists. A total of 8557 entries were arranged for the RT-PCR test(s). After the exclusion of entries outside the scope of the study (uncertain contact history with various reasons for the RT-PCR test), duplicate entries and those without sufficient data for analysis, 7146 exposures were retained in the final dataset.

### 3.1. Baseline Characteristics

Of the 7146 exposures of 5449 hospital workers, 299 (4.2%) cases of COVID-19 infection were confirmed. The incidence of included events and COVID-19 detection gradually decreased during the study period (Appendix A). The baseline characteristics of the included entries are listed in Table 1. The median age (range) of exposed hospital workers was 32 years (18–88), with women (73.8%) and healthcare personnel (Appendix B, 85.6%) being predominant. Among the hospital workers, the most common occupations were nurses and nurse/physician assistants (41.1%) followed by physicians/dentists and dentist assistants (12.6%), janitorial staff (12.3%), and administrative staff (12.3%). Less than 1% of the entries came from hospital workers with previous COVID-19 disease, and no hospital worker experienced repeated infection during the study period. In general, SARS-CoV-2 detections were more prevalent in exposures of workers with lower education (primary or secondary school; 7.7%), exposures without proper personal protective equipment or hygiene (i.e., high-risk exposure; 8.1%), exposures accompanied by fever or other symptoms related to COVID-19 (Appendix B, 14.3%), and exposures of hospital workers who had received vaccine combinations of lower potency (low potency; 14%).

All events were classified into four exposure risk categories: low (31.9%), moderate (24.1%), high (38.9%), and insignificant risk (but being tested due to COVID-19-related symptoms) (5.1%). This risk classification was highly correlated with the SARS-CoV-2 detection rate (0.7%, 2.3%, 8.1%, and 5.3%; *p* < 0.001). Most exposures (98.1%) came from personnel who had received at least one dose of the vaccine. The median interval from the last vaccination to the day of exposure was 72 days (range 14 to 236). More than half of the hospital workers (54.2%) received two doses of CoronaVac (SINOVAC Biotech, Beijing, China), 17.5% received an additional ChAdOx-1 (AstraZeneca, Oxford, UK; Cambridge, UK), 15.2% received an additional BNT162b2 (Pfizer-BioNTech, New York, USA; Mainz, Germany) vaccination as a booster, and 11.2% had other vaccine combinations. The remaining 136 exposures came from hospital workers who were not vaccinated at the time of exposure (1.9%).

Among the events with subsequent COVID-19 infection, the median time to detection after the last exposure was four days (interquartile range 1 to 7), with 90% of all detections occurring within 11 days from the last exposure (Appendix A). No mortality was observed during the study period.

### 3.2. Factors Associated with SARS-CoV-2 Infection

After prescreening with univariate logistic regression, twelve factors entered the preliminary main effect model (Table 2), and nine remained in the final logistic model. There were two baseline characteristics and seven exposure-related factors that contributed to the risk of SARS-CoV-2 infection. All independent factors and weights associated with them are shown in Table 3. To calculate the predicted probability for SARS-CoV-2 genetic detection using an additive risk score, the points for factors present in a particular exposure are added to give an approximate percentage, as outlined in Table 4.

Having a fever or other COVID-19-related symptoms was the strongest risk factor for SARS-CoV-2 genetic detection (adjusted OR 4.94, 95% CI 3.83–6.39). Other strong risk factors included performing an aerosol-generating procedure without full protection (aOR 2.87, 1.66–4.96), prolonged duration of contact (aOR 2.52, 1.82–3.49), and personnel not wearing a mask (aOR 2.49, 1.75–3.54). Direct contact with aerodigestive secretion, the infected person not wearing a mask, and close contact without proper eye protection carried smaller risks. Vaccination was protective against infection: aOR 0.05 (high-potency combinations), aOR 0.17 (moderate-potency combinations), and 0.3 (low-potency combination). Hospital workers with higher levels of education level were less likely to be infected.

The model fit was confirmed using the Hosmer and Lemeshow test (Chi-square 8.960, *p* 0.346). The discrimination power of the final logistic model and the risk scoring system accessed via ROC curves are depicted in Figure 2, which confirms the model’s performance. The exposure risk categories also demonstrated good predictive power in the parallel analysis (adjusted OR 2.58 for moderate-risk and 8.53 for high-risk contact; Appendix A), but with a smaller area under the ROC curve at 0.827 (95% CI 0.804–0.849).

## 4. Discussion

Using information acquired from contact tracing during the Delta peak at 86–99% in the community [20,21,22], we developed a risk prediction model to estimate the risk of infection for hospital workers with different vaccination regimens following exposure to confirmed COVID-19 cases. Exposure type, the presence of symptoms, the appropriate use of PPE, education level, vaccination regimen, and time since the last dose each contributed important information regarding the risk of infection.

Having a fever or other COVID-19-related symptoms within two weeks was strongly predictive of a positive test. Similar to the previous report by Pienthong et al. [8], failure to comply with protective measures increased the risk of infection. For example, commencing an aerosol-generating procedure (Appendix B) without proper protective equipment (including an N95 respirator and eye protection) was the highest procedural risk in this study, followed closely by a prolonged duration of exposure and the worker not wearing a mask. Other violations of standard precautions and the improper use of PPE recommended by the WHO [23] also increased the risk of infection. One interesting finding to be noted is that an exposure distance of <1 m and not using an eye protection device failed to reach statistical significance in the preliminary effect model but showed significance when considering both factors together (i.e., a face shield is only beneficial when in close contact). This supports the adequacy of the universal droplet precautions despite recent evidence in favor of airborne precautions [24,25] given that no aerosol-generating procedure is being performed.

The most common vaccine regimen in this study, two doses of inactivated vaccines (low potency), provided the least protection against infection, while the second most common regimen, heterologous boosted inactivated vaccines (moderate potency), provided slightly better protection but much less when compared with the viral vector-mRNA combination (high potency). This is consistent with the previous report from Sritipsukho et al. [17] which underlined the importance of vaccine type over the number of doses. Our findings also validated our COVID-19 exposure risk category approach which was used to determine the need for RT-PCR testing and isolation during a period of manpower and resource limitation.

Although symptoms related to COVID-19 should be considered as a consequence of infection rather than a risk factor for infection, our data support that all symptomatic health workers with an exposure history during the epidemic should be tested, regardless of contact risk and immunologic status, provided that this policy does not overwhelm laboratory testing capacity. A significant portion of infected hospital workers tested positive before the initial recommended test date(s), which implied the benefit of the early test (and early detection) triggered by symptoms. This contrasts with other studies on symptomatic patients presenting at health services which demonstrated poor diagnostic accuracy of signs and symptoms [26,27]. An explanation might be that, in addition to being symptomatic, all of our included subjects must have certain exposure to an infected person.

Consistent with a 2020 study by Chadeau-Hyam et al., the level of education of the hospital workers was inversely correlated with the risk of testing positive [28]. This could be explained by better health literacy, self-awareness, and hygiene discipline. Educational achievement is also correlated with occupations that pose different risks of COVID-19 infection [29]. Improved educational interventions are additionally needed to increase awareness among workers with lower levels of education.

Most of the COVID-19 risk calculators available provide a very crude risk estimate based primarily on location, the nature of the activity, and the safety measures being taken [30]. Our risk calculator and score, on the contrary, provide an individualized risk assessment based on detailed exposure characteristics adjusted for vaccination status and socioeconomic background through educational attainment. To a certain extent, this tool has the utility to triage exposed individuals to prevent further infections in healthcare settings.

This study has several limitations. We did not include the severity of cases that got infected (i.e., CT value or hospitalization). Due to the retrospective nature of the observational study, some demographic information may have been missed. Furthermore, most of the data were entered by various staff with different levels of health knowledge. Therefore, misclassification may be an issue. The external validation of the risk model was also difficult to perform due to the rapid shift in the variants of concern and vaccine-induced immunity over time.

## 5. Conclusions

Having symptoms of COVID-19, inadequate personal protection, low education level, and not receiving a vaccine or receiving a low-potency vaccine regimen were found to be the main risks for COVID-19 infection among all healthcare-related exposures. Our quantitative exposure risk model and risk score have good predictive value and could help combat further spread among hospital workers according to their actual probability of infection.

## Figures and Tables

**Figure 1 tropicalmed-07-00248-f001:**
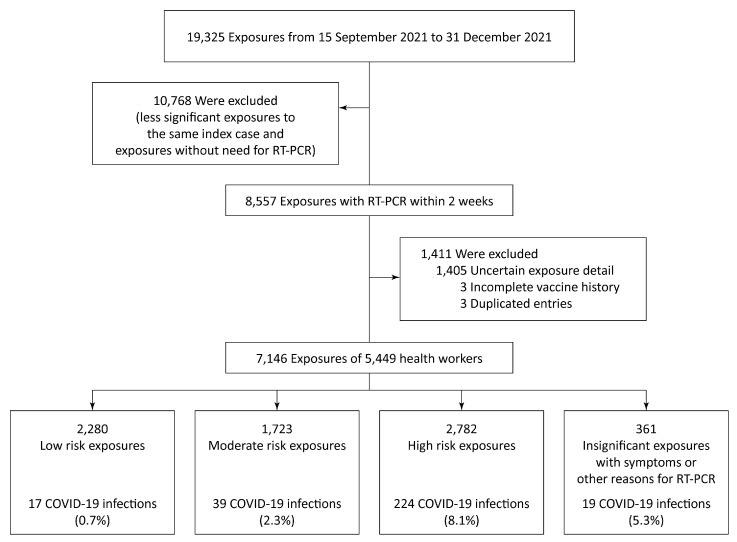
Consort type study flow diagram.

**Figure 2 tropicalmed-07-00248-f002:**
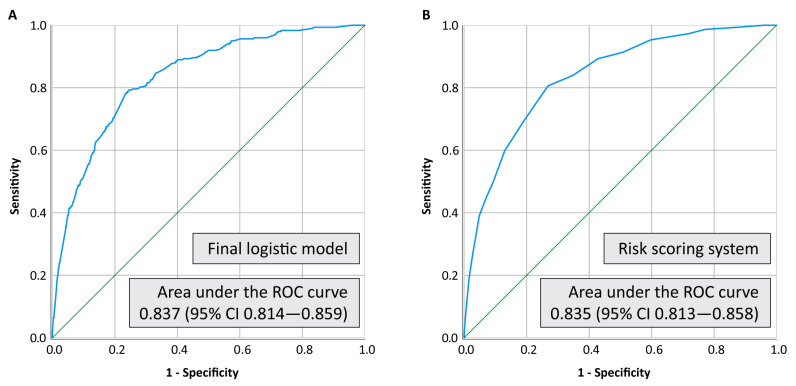
Areas under the ROC curve for the final logistic model (**A**) and the risk scoring system (**B**).

**Table 1 tropicalmed-07-00248-t001:** Characteristics of occupational exposures to COVID-19 of hospital workers.

Characteristics	Subsequent COVID-19 Infection within 14 Days after Last Exposure			Total		*p* Value
	No	Yes				
	*n* = 6847	*n* = 299	Event Rate	*n* = 7146	% of Total	
**Demographic**						
Age at exposure, year						
Mean, standard deviation	34.95, 10.49	35.72, 10.64		34.98, 10.50		0.216
Median (interquartile range)	32 (27–42)	35 (26–44)		32 (27–42)		0.186
Gender						0.067
Male	1781	92	4.9%	1873	26.2%	
Female	5066	207	3.9%	5273	73.8%	
The highest education attainment						<0.001 ^§^
Primary or secondary school	1599	133	7.7%	1732	24.2%	
Associate’s degree	1296	69	5.1%	1365	19.1%	
Bachelor’s degree	2846	80	2.7%	2926	40.9%	
Master’s degree	762	12	1.6%	774	10.8%	
Doctoral degree	344	5	1.4%	349	4.9%	
Role of hospital worker						0.620
Healthcare personnel	5864	253	4.1%	6117	86.6%	
Non-healthcare personnel	983	46	4.5%	1029	14.4%	
**COVID-19 vaccination status**						
Vaccines						<0.001
CoronaVac–CoronaVac	3684	190	4.9%	3874	54.2%	
CoronaVac–CoronaVac–ChAdOx-1	1203	47	3.8%	1250	17.5%	
CoronaVac–CoronaVac–BNT162b2	1070	18	1.7%	1088	15.2%	
ChAdOx-1	284	10	3.4%	294	4.1%	
ChAdOx-1–ChAdOx-1	219	9	3.9%	228	3.2%	
None	117	19	14.0%	136	1.9%	
ChAdOx-1–BNT162b2	116	1	0.9%	117	1.6%	
Others	154	5	3.1%	159	2.2%	
Potency of COVID-19 Vaccines *						<0.001 ^§^
None	117	19	14.0%	136	1.9%	
Low-potency vaccines	4025	202	4.8%	4227	59.2%	
Moderate-potency vaccines	2537	77	2.9%	2614	37.6%	
High-potency vaccines	168	1	0.6%	169	2.4%	
The interval between the last dose of COVID-19 vaccines and exposure, day						
Mean, standard deviation	72.07, 33.36	73.78, 29.68		72.14, 33.22		0.351
Median (interquartile range)	72 (47–93)	75 (57–95)		72 (48–93)		0.302
Missing data	207	21		228	3.2%	
**Previous COVID-19 infection**						0.755 #
Absence	6564	290	4.2%	6854	99.1%	
Presence	62	3	4.6%	65	0.9%	
**Exposure characteristics**						
Infected person was wearing a mask/N95 respirator during exposure						<0.001
Yes	2897	61	2.1%	2958	41.4%	
No	3950	238	5.7%	4188	58.6%	
Distance of contact						<0.001
More than 1 m	1510	40	2.6%	1550	21.7%	
Less than 1 m	5337	259	4.6%	5596	78.3%	
Duration of exposure						<0.001
Less than 15 min	3380	53	1.5%	3433	48.0%	
More than 15 min	3467	246	6.6%	3713	52.0%	
Exposed hospital worker was wearing a mask/N95 respirator during exposure						<0.001
Yes	4535	91	2.0%	4626	64.7%	
No	2312	208	8.3%	2520	35.3%	
Exposed hospital worker was wearing a face shield during exposure						<0.001
Yes	1941	38	1.9%	1979	27.7%	
No	4906	261	5.1%	5167	72.3%	
Infected person was undergoing aerosol-generating procedures						0.186
No	6465	277	4.1%	6742	94.3%	
Yes; exposed hospital worker was wearing N95 respirator/PAPR and face shield	77	2	2.5%	79	1.1%	
Yes; exposed hospital worker was not wearing N95 respirator/PAPR and face shield	305	20	6.2%	325	4.5%	
Exposed hospital worker had direct contact with the aerodigestive secretion of the infected person						<0.001
No	6549	249	3.7%	6798	95.1%	
Yes	298	50	14%	348	4.9%	
Exposure risk category by infectious disease physicians						<0.001
Low risk	2263	17	0.7%	2280	31.9%	
Moderate risk	1684	39	2.3%	1723	24.1%	
High risk	2558	224	8.1%	2782	38.9%	
Insignificant exposure with symptom(s) or reason(s) for RT-PCR	342	19	5.3%	361	5.1%	
**Symptom of exposed hospital worker**						
Fever or other COVID-19-related symptoms						<0.001
Absence	5073	103	2.0%	5176	79.1%	
Presence	1174	196	14.3%	1370	20.9%	

RT-PCR; reverse transcriptase–polymerase chain reaction, § linear-by-linear association, # Fisher’s Exact test, other *p* value from independent samples *T*-test, Pearson Chi-Square test, or independent-samples Mann–Whitney U test, * adapted from Thai COVID-19 Vaccination Guidelines for a Booster Shot, Ministry of Public Health, December 2021.

**Table 2 tropicalmed-07-00248-t002:** Logistic regression analysis of variables associated with occupational SARS-CoV-2 infection among hospital workers.

Variable	Univariable Analysis			Multivariable Analysis		
	Crude OR	(95% CI)	*p* Value	Adjusted OR	(95% CI)	*p* Value
**Demographic**						
Age (year)	1.01	(1–1.02)	0.216	1.01	(1–1.02)	0.053
Male gender	1.26	(0.98–1.63)	0.068	1.11	(0.83–1.48)	0.480
The highest education attainment			<0.001			<0.001
Primary or secondary school (reference)						
Associate’s	0.64	(0.47–0.86)	0.004	0.76	(0.54–1.06)	0.106
Bachelor’s	0.34	(0.25–0.45)	<0.001	0.44	(0.32–0.61)	<0.001
Master’s	0.19	(0.1–0.34)	<0.001	0.31	(0.17–0.58)	<0.001
Doctoral	0.18	(0.07–0.43)	<0.001	0.36	(0.14–0.92)	0.033
Role of worker: Healthcare personnel	0.92	(0.67–1.27)	0.620			
**Exposure characteristics**						
Infected person was not wearing a mask/N95 respirator during exposure	2.86	(2.15–3.81)	<0.001	1.45	(1–2.1)	0.048
Distance of exposure less than 1 m	1.83	(1.31–2.57)	<0.001	1.4	(0.97–2)	0.069
Duration of exposure more than 15 min	4.53	(3.35–6.11)	<0.001	2.51	(1.81–3.48)	<0.001
Exposed hospital worker not wearing a mask/N95 respirator during exposure	4.48	(3.49–5.77)	<0.001	2.54	(1.72–3.76)	<0.001
Exposed hospital worker not wearing face shield or goggles during exposure	2.72	(1.93–3.83)	<0.001	1.25	(0.78–1.98)	0.353
Infected person was undergoing aerosol-generating procedures			0.156			0.001
No (reference)						
Yes; exposed HCP was wearing N95 respirator/PAPR and face shield	0.61	(0.15–2.48)	0.486	1.28	(0.29–5.66)	0.748
Yes; exposed HCP was not wearing N95 respirator/PAPR and face shield	1.53	(0.96–2.44)	0.075	2.86	(1.64–5)	<0.001
Exposed hospital worker had direct contact with aerodigestive secretion of the infected person	4.41	(3.19–6.11)	<0.001	1.48	(1.02–2.15)	0.038
**Symptoms of exposed hospital worker**						
Fever or other COVID-19-related symptoms	5.44	(4.26–6.95)	<0.001	4.9	(3.78–6.34)	<0.001
**COVID-19 vaccination status**						
Potency of COVID-19 vaccines *			<0.001			<0.001
None (reference)						
Low-potency vaccines	0.31	(0.19–0.51)	<0.001	0.31	(0.18–0.54)	<0.001
Moderate-potency vaccines	0.19	(0.11–0.32)	<0.001	0.16	(0.09–0.3)	<0.001
High-potency vaccines	0.04	(0.01–0.28)	0.001	0.05	(0.01–0.41)	0.005
The interval between the last dose of COVID-19 vaccines and exposure (day)		(1–1.01)	0.402			
**Previous COVID-19 infection: Yes**	1.1	(0.34–3.51)	0.878			

* Adapted from Thai COVID-19 Vaccination Guidelines for a Booster Shot, Ministry of Public Health, December 2021.

**Table 3 tropicalmed-07-00248-t003:** Independent risk factors associated with subsequent SARS-CoV-2 infection after occupational exposure among hospital workers, coefficients from the final logistic model, and weight (point) for the risk score.

Risk Factor	β	Odds Ratio (95% CI)	*p* Value	Point
The highest education attainment			<0.001	
Primary or secondary school (reference)				3
Undergraduate (associate’s or bachelor’s)	−0.64	0.53 (0.4–0.68)	<0.001	1
Postgraduate (master’s or doctoral)	−1.13	0.32 (0.19–0.55)	<0.001	0
Infected person was not wearing a mask/N95 respirator during exposure	0.37	1.44 (1.01–2.07)	0.046	1
Distance of exposure less than 1 m without a face shield	0.33	1.39 (1.02–1.89)	0.038	1
Duration of exposure more than 15 min	0.93	2.52 (1.82–3.49)	<0.001	3
Exposed hospital worker was not wearing a mask/N95 respirator during exposure	0.91	2.49 (1.75–3.54)	<0.001	3
Exposed hospital worker was not wearing an N95 respirator and face shield/goggles while the infected person was undergoing aerosol-generating procedure	1.05	2.87 (1.66–4.96)	<0.001	3
Exposed hospital worker had direct contact with the aerodigestive secretion of the infected person	0.40	1.5 (1.03–2.17)	0.033	1
Fever or other COVID-19-related symptoms	1.60	4.94 (3.83–6.39)	<0.001	5
Potency of COVID-19 vaccines *			<0.001	
None (reference)				9
Low-potency vaccines	−1.19	0.3 (0.17–0.53)	<0.001	5
Moderate-potency vaccines	−1.79	0.17 (0.09–0.3)	<0.001	4
High-potency vaccines	−2.98	0.05 (0.01–0.4)	0.004	0
Constant	−3.69		<0.001	

* Adapted from Thai COVID-19 Vaccination Guidelines for a Booster Shot, Ministry of Public Health, December 2021.

**Table 4 tropicalmed-07-00248-t004:** The predictive score for SARS-CoV-2 infection after occupational exposure among hospital workers.

Total Point	Predicted Probability of COVID-19 Infection (%)
0–9	0.05–0.93
10–14	1.28–4.60
15–16	6.28–8.51
17–19	11.44–19.94
20–23	25.70–48.09
24–29	56.27–86.92

## Data Availability

The datasets generated during and/or analyzed during the current study are available from the corresponding author upon reasonable request.

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
