# Peer review of "A Risk Prediction Model and Risk Score of SARS-CoV-2 Infection Following Healthcare-Related Exposure"

_tropicalmed, 2022, doi:10.3390/tropicalmed7090248_

Round 1

Reviewer 1 Report

Dear authors, 

This is a very interesting article about developing a risk score for SARS-CoV2 infection.

As it is, it is worth publishing provided some minor changes are done.

- line 70 - the authors should specifiy up to which moment was the data gathered. It is also important to specify if data gathering only related to the delta variant, or if omicron was also included. If it was included, maybe data should be separated depending on the strain; or some temporal trends should be evaluated

- line 162 - "These risk classifications" should be at singular

- line 163 - "Most of the exposures" should be "Most exposures"

- line 188 - the logistic exposure risk calculator info should be added at the materials and methods, not the results section

- I would like to see a more expanded discussion section. There is enough data in the scientific literature to compare. Moreover, the number of references is quite low.

Reviewer 2 Report

Article

A risk prediction model and risk score of SARS-CoV-2 infection following healthcare-related exposure

This study is aimed to identify factors associated with SARS-CoV-2 infection in hospital workers with known exposure(s) to confirmed positive patients and to build an evidence-based risk model and risk score for healthcare-related exposure.

In fact this study answered the research question but many comments are there.

-          It contains some of typing and grammatical mistakes

-          It is well written and designed

-          Objectives need some clarification to be clear.

-          How the authors determine the risk score of SARS-CoV-2 infection, do they have followed some guidelines, or they just followed previous studies, please provide reference.

General points to be considered

v  Title: perfect

-          Abstract: perfect.  

1.      Introduction:

Well written

In the introduction lines 65-67, the objective needs to be rewritten perfectly. Also it should be corrected.

The objectives of this study are to identify factors associated with SARS-CoV-2 infection in hospital workers with known exposure(s) to confirmed positive patients and to build an evidence-based risk model and risk score for healthcare-related exposure.

2.      Materials and Methods: well written.

2.1  Study design, setting, and protocol

In line 70: the study period should be determined accurately, from….to……

This study is a retrospective cohort analysis. From July 2021 to………., during the increase in the number of cases of COVID-19 caused by the Delta variant, the hospital implemented

Statistical analysis: Well written

Results:

In 172; this sign should be removed.

RT-PCR; Reverse transcriptase-polymerase chain reaction, independent-samples Mann-Whitney U test, § linear-by-

Discussion: well written, but should be English language writing should be checked.

Conclusion: well written, but could be improved.

v  References: citation of references should be according to journal instructions.

v  Year of study should be written in bold.

v  References should be checked and standardized.
